# Orbital Polarization-Dependent Fragment Twist-Induced Intramolecular Electric-Field-Driven Charge Transfer

**DOI:** 10.3390/molecules28041801

**Published:** 2023-02-14

**Authors:** Wenjing Bo, Hao Sheng, Jingang Wang

**Affiliations:** Liaoning Provincial Key Laboratory of Novel Micro-Nano Functional Materials, College of Science, Liaoning Petrochemical University, Fushun 113001, China

**Keywords:** intramolecular electric field, charge transfer, optical absorption, electron circular dichroism

## Abstract

Defects, such as twisting, in fused aromatic hydrocarbons disrupt the plane of the π orbital. The twisted structure induces an electric field in the system and affects the spectra. In this work, theoretical studies show that the intramolecular electric field within a distinctly twisted structure is larger than that of other molecules. In addition, the spectral study shows that the degree of charge transfer and the magnetic transition dipole in the electrostatic potential extremum region of the molecular electric field were significantly improved, which affected the optical absorption and chiral optical behavior of the molecule. The discovery of this theoretical regulation law will provide a solid foundation for the electric-field-induced regulation of optical properties and will promote the precise design and synthesis of optoelectronic molecules with inner electric fields.

## 1. Introduction

The intramolecular electric field is a powerful tool for the design of optoelectronic materials and catalysts [1,2,3]. Theoretical and experimental research into intramolecular electric fields are very important in many fields. For example, when applied to solar cells, Yang et al. found that the electron transmission in molecular local electric fields is critical for improving the performance of perovskite solar cells [4,5,6,7]. In terms of the molecular electric-field regulation, Mu et al. reviewed the molecular electric-field regulation of porphyrin/phthalocyanine photoelectric materials and provided a new research idea for the design of such materials [8,9]. Xie et al. found that new porphyrin compounds with a tunable structure can be synthesized by twisting the molecular structure [10,11,12]. Cao et al. made a major breakthrough in the artificial regulation of small-molecule activation and catalytic conversion involving proton transfer [13,14,15]. In terms of bioluminescence, Tang et al. proposed an aggregation-induced luminescence (AIE) technology based on molecular distortion that is widely used in biological probes, cell imaging, diagnosis, and treatment [16]. The molecular adjustment mechanism used in this work [17,18,19,20], in order to adjust the optical properties, is not an electronic transition caused by external factors, but an electronic transition caused by the internal electric field generated by molecular distortion.

Twisted and fused aromatic hydrocarbons recombine into new types of nanographene. Nanographene is graphene with a size of the nanometer scale. Graphene is a two-dimensional nanomaterial, which has a high carrier mobility and high performance in optics, and has been widely studied in physics, chemistry, and other disciplines [21,22,23,24,25,26]. Nanographene composites can be tuned by size effect and shape effect. Adjustment of its width, boundary, defects, doping of other atoms, etc., leads to changes in the electronic structure and electrical properties of nanographene. For example, the zigzag edge (ZZ) and armchair type (AC) exhibit different electronic and optical properties. Because of the unique properties of nanographenes, they can be widely used in nanotransistors, solar cells, optoelectronics, etc.

Owing to the internal twisting of the nanographene, an intramolecular electric field is generated, which may regulate its optical absorption properties. Therefore, in order to investigate the regulation mechanism of the intramolecular electric field on the optical absorption properties, six nanographene molecules with similar geometric structures, produced by the oxidative rearrangement of helicene, were studied in this paper. This work first studies the polarity of the molecule by establishing the fragment dipole moment, but it is not enough to measure the polarity of the molecule using only the dipole moment. The electrostatic potential is also determined, and it is found that molecules with twisted fragments are prone to electrophilic effects, which induce molecules to generate intramolecular electric fields. An electrophilic reaction is a reaction caused by an electron-deficient (high electron-affinity) reagent attacking a region in another compound with a higher electron cloud density (electron-rich). The electrostatic potential and fragment dipole moments demonstrate the generation of the intramolecular electric field. The molecular orbitals and electron holes are then determined, and it can be concluded that the generation of an intramolecular electric field causes orbital polarization, which causes the molecule to exhibit light-absorbing characteristics.

## 2. Results and Discussion

### 2.1. Inner Electric Field of Nanographene

A change in the molecular geometry will cause a change in its spatial structure of nanographene, known as the distortion phenomenon, which induces the generation of an electric field inside the molecule. The dipole moment can be regarded as the local internal electric field generated by the electric dipole; therefore, in this section, we use the fragment dipole moment [27] as a means to calculate the internal electric field of the system. It can be observed that there are six, neatly arranged benzene rings on the left side of the six molecules; therefore, the whole system can be regarded as the fusion of the benzo[ghi]perylene moiety (six benzene ring moieties) with the helicene skeleton, in order to calculate the fragment dipole moments. As shown in Figure 1a, taking O7H as an example, the left side of the dotted line is the benzopyrene portion, and the right side of the dotted line is the remaining helicene skeleton.

After calculation, the dipole moment and the dipole moment components of the benzo[ghi]perylene side of the compound and the whole system are shown in Appendix A. We all know that the larger the dipole moment of the molecule, the greater the polarity of the molecule, and vice versa. It can be seen from the data in Appendix A that larger π-extension behavior leads to the deformation of the structure, making the distance between the positive and negative charge centers larger. It can be seen that the dipole moment of the corresponding O9H is larger, and that the corresponding polarity is also larger. The direction of the dipole moment is from the negative electric center to the positive electric center. Looking at O9H (Figure 1c), the starting point of the blue arrow is located at the junction of the helical olefin skeleton and the benzopyrene structure. The results show that the twisted segment of the helicene is negatively charged, and that the charge transfer occurs. Compared with the dipole moment of other molecules, the dipole moment of the twisted fragment of O9H is larger, resulting in a larger intramolecular electric field. Distorted changes in the spatial structure of molecules can induce the generation of intramolecular electric fields, and intramolecular electric fields within obviously distorted structures are larger than those of other molecules.

To investigate whether the intramolecular electric field induced by the fragment twist has an effect on the optical properties, the UV–Vis spectra of the molecules was analyzed. From Figure 1g,h and Appendix A, we can see that their wavelength changes in the visible wavelength range (390~780 nm). Comparing the absorption spectra of the O7H, O8H, and O9H molecules (orange, red, and violet curves), the wavelengths are gradually red-shifted. The absorption peak of O7H is at 413.11 nm, which is contributed by S_1_. The absorption peak of O8H is at 422.66 nm, which is contributed by S_1_. The absorption peak of O9H is at 440.26 nm, which is also contributed by S_1_. It can be inferred that as the degree of molecular distortion increases, the electric field in the molecule increases, and the optical properties of the molecule increase.

### 2.2. Electrostatic Potential

The intramolecular electric field related to orbital polarization will affect the charge transfer behavior. The intramolecular electric field can produce opposite forces on charges with different electrical properties in the molecule, promoting charge separation and transport and resulting in an uneven distribution of the charge density in the molecule. Analysis of the electrostatic potential demonstrates that the intramolecular electric field that is generated by molecules with twisted structures promotes the charge separation and transport [28,29,30,31,32,33], resulting in differences in the electrostatic potential between the different molecules. The molecular polarity parameters also confirm this phenomenon.

Figure 2 shows the surface electrostatic potentials of six molecules, represented in blue, white, and red, respectively, and the electrostatic potential diagram (Appendix A) was analyzed from multiple perspectives. The electrostatic potential value of the red area is positive, indicating that this region more easily donates electrons, or, is more nucleophilic than the other regions; whereas the electrostatic potential value of the blue region is negative, indicating that this region is more likely to obtain electrons or is more electrophilic than the other regions. It can be clearly seen that the positive electrostatic potential extremum region and the negative electrostatic potential extremum region are both distributed on the surface of the molecule, but the difference is in the location of their distribution. First, comparing the electrostatic potential of each molecule, it is evident that the larger the twisted molecule, the larger the difference between the extreme positive and extreme negative electrostatic potential of the molecule. This indicates that the electric field within the molecule has a greater effect on charges with different electrical properties. The extreme negative value of O9H is −27.31 kcal/mol, which is larger than those of the other five molecules (Appendix A). Second, it is evident that the degree of distortion of the six intramolecular structures is different, so that the extreme negative electrostatic potential is located on the molecular surface at the molecular twist, indicating that the intramolecular electric field generated by twisting of the molecular structure separates the electrons and moves them to the molecular surface. It can be demonstrated by the result obtained in Section 2.1 that the π-expansion behavior makes the structure more distorted; the distance between the positive and negative charge centers becomes larger, the dipole moment of O9H becomes larger, the corresponding polarity becomes larger, and finally, produces an intramolecular electric field that is also larger. Anyway, it is clear that the electrostatic potential extremum region promotes electron transport. The MPI index of O9H (10.60 kcal/mol) was also calculated and Appendix A shows that it is larger than the MPI index of the O7H and O8H molecules, indicating that O9H has the larger polarity and the larger intramolecular electric field [34,35].

### 2.3. Molecular Orbital Analysis

Orbital polarization-dependent fragment twist induces the generation of intramolecular electric fields. The purpose of molecular orbital analysis in this section is to better study the electronic transitions caused by changes in molecular spatial structure, in order to observe the distribution of orbital energy levels, and then describe the optical properties. By drawing the density of states (DOS) maps [36,37,38,39,40], the orbital characteristics of different energy ranges can be clearly seen. Figure 3 shows the electron cloud distribution of the molecular orbital of O9H, in which HOMO(H) and LUMO(L) are the main orbitals that play an important role in molecular stability. From the molecular orbital point of view, the contribution of H to L in the S_1_ state is 93.4%; the distribution of H and L electron clouds is roughly the same, but the color is roughly opposite, indicating that S_0_→S_1_ is a charge transfer behavior, which is an obvious transition feature. Looking at the other orbitals, the electron cloud at the twist of H-1 is dense, corresponding to the extremely negative electrostatic potential of O9H. The H-5 and H electron clouds are mainly concentrated in the benzopyrene region on the left, and their electron cloud density is significantly different, whereas the L+1 electron cloud is mainly concentrated in the helicene region on the right. It is evident that charge transfer behavior occurs in the molecule. The orbital energy levels and H-L energy level differences of the six molecules were also calculated (Appendix A). Their optoelectronic properties can be determined from the results. The energy level of the H orbital rises significantly, as seen in O9H, OO8H, and OO9H. Their common feature is that the tail is helical and not coplanar with the benzene ring at the front part. The energy gap order of the H-L orbital is OO9H < OO8H < O9H < O8H < O7H < OO7H. The H energy of O9H is −6.2 eV, the L energy is −1.2 eV, and the energy difference between H and L is the smallest (4.98 eV), indicating that the conductivity is also better than that of O7H and O8H. For this fused benzene ring derivative, its H-L orbital energy gap decreases with the increase in the conjugated system, so that the molecule has good electrical and optical properties, which can be used in superconducting materials, semiconductor materials, and optoelectronic materials, etc.

For such close structures, they have similar energies and roughly the same number of states. In the energy level range, the overall density of states is dominated by the $P_{z}$ orbitals, which allows more electrons to remain in the $P_{z}$ orbitals and be spin-polarized, which in turn enhances the local spin-polarization. The DOS between the two spikes flanking the Fermi level of O9H in the six figures is not zero and is broad. It shows that the covalent bond of the system is relatively strong at this time, and that the smaller the electronegativity, the greater the polarization. As shown in Figure 4, in the energy level range from approximately −21.7 to −5.3 eV, the total density of states is dominated jointly by $P_{x}$, $P_{y}$, and $P_{z}$ orbitals. In the energy level range from around −21 to −10 eV, the total density of states is dominated by $P_{x}$, $P_{y}$, and $P_{z}$ orbitals; whereas, in the energy level range from around −10 to 5.5 eV, the total density of states is mainly dominated by the $P_{z}$ orbital [41], which indicates that the orbital polarization is caused by the $P_{z}$ orbital.

### 2.4. One Photon Electron-Hole Map (OPA) and Transition Density Matrix (TDM)

In this section, we study the effect of orbital polarization-dependent electric-field modulation on electron transition behavior in molecules, which in turn, affects the optical properties of molecules. Figure 5 shows an electron-hole diagram. For the electron-hole density diagram in Figure 5, the blue and red isosurfaces describe the electron and hole density, respectively. The diagonal elements of S_1_, S_6,_ and S_12_ of O9H have little change and display the charge transfer behaviors. The data in Appendix A are consistent with these results, showing the charge transfer behavior within the molecule. By analyzing the isosurface, it is evident that this charge transfer behavior comes from the distorted region of the fragment within the molecule, which reflects the charge transfer behavior of the twisted region. The intramolecular electric field is generated through induction from the twisting of the structure, which makes the electrostatic potential inside the molecule extremely negative, so that the positively charged holes in the isosurface map are transferred to the twisted structure. It shows that the twisted structure plays the role of the electron donor, and the benzo[ghi]perylene part plays the role of the electron acceptor. Since the twisted structure affects the molecular orbital wave function, an electric field is generated within the molecule, resulting in a change in the corresponding transition energy (absorption wavelength), which in turn, affects the optical properties of the molecule.

The electron-hole analysis diagram, the transition density analysis diagram, the degree of separation of electrons and holes, and the degree of polarization of the intramolecular electric field were analyzed in combination with the characteristic parameters of the main electron excitation state of the O9H molecule in Appendix A. Most of the compounds in the table have a relatively large $S_{r}$ and $S_{m}$ index, indicating a high degree of overlap between the holes and electrons. However, for O9H, these two indexes are relatively small (for example, 0.86917 and 0.65921 of S_1_), indicating that the separation of the holes and electrons is more obvious. It can also be clearly observed in the transition density map that the holes and electrons of other compounds have a high degree of overlap (when the index is close to one, it means that the holes and electrons are perfectly coincident, and vice versa), and the charge transfer is obvious. The t indexes in the table are all less than zero, which means that the separation of the holes and electrons is not obvious in the direction of the charge transfer excitation. When the D index in the table is large (for example, 0.263 of S_1_), it indicates that the excitation is the charge transfer and that the distance between the electrons and holes is large, which is consistent with the above description. In addition, the isosurface of the twisted segment becomes larger, indicating more electron transfer.

### 2.5. Transition Electric Dipole Moment (TEDM) and Transition Magnetic Dipole Moment (TMDM)

In this section, it was found that the intramolecular electric field is generated due to the fragment twist, resulting in significant differences between the transition electric couple and the magnetic couple. Figure 6 shows the three-dimensional and two-dimensional transition dipole and magnetic dipole moment densities of S_1_ of O9H. As can be seen from Figure 6, it can be considered that the S_1_ transition electric dipole moment and the transition magnetic dipole moment density of O9H are roughly symmetrical along the X and Y direction, and the symmetry axes are orthogonal to each other. However, the aggregation density on both sides of the symmetry axis is different, and TEDM and TMDM are complementary in area. With regard to the Z direction, TEDM and TMDM can be regarded as complementary in area, but they are not symmetrically distributed along the Z direction. The Z direction of the TMDM isosurface map of O9H is mostly yellow. The isosurface density of the Z direction of TEDM is mainly concentrated in the twisted region, and there are many purple regions. In addition, S_1_ of O7H and S_1_ of O8H (Appendix A) in the X and Y directions are similar to S_1_ of O9H, the difference is in the Z direction. The Z direction of the TMDM isosurface of S_1_ of O7H is mostly blue, and the Z direction of the TEDM isosurface is mostly yellow. The Z direction isosurface of S1 of O8H is roughly the same as that of S1 of O7H. The electronic circular dichroism (ECD) of the excited states of the above three molecules are all positive. Among them, the ECD peak position of the S_1_ excited state of O9H is the highest, indicating that the electromagnetic interaction of the S_1_ excited state of O9H is stronger than that of the S_1_ excited state of O7H and the S_1_ excited state of O8H. According to the tensor product between the transition electric dipole moment and the transition magnetic dipole moment, S_1_ of the O9H molecule will show a stronger rotor strength than S_1_ of the O7H molecule and the S_1_ of O8H, indicating that theECD of S_1_ of the O9H molecule has a larger peak, which can be seen in the ECD spectrum of Appendix A. 

Finally, it is evident that structural distortion has a substantial effect on the chiral electromagnetic interactions, especially for TMDM. The distribution of the TMDM, X, Y, and Z components of S_3_ of OO8H and S_5_ of OO9 H in Figure 7 is roughly the same as that for the above three molecules. The difference is the color distribution on the isosurface. It can be concluded that the color distribution law of the TMDM isosurface is the opposite (the blue isosurface is positive and the yellow isosurface is negative), which will lead to opposite absorption peaks of the ECD spectrum. The chirality of the molecule is reversed.

### 2.6. Circular Dichroism

By analyzing the electron circular dichroism, it can be proved that the orbital polarization-dependent intramolecular electric field affects the optical properties of molecular chirality by affecting the transition behavior of electric and magnetic couples [42,43]. When linearly polarized light passes through an optically active substance, the phenomenon of deflection is called the Cotton effect. Comparing the calculated ECD spectrum with the experimental ECD spectrum, it can be seen that, although the calculated band position has a certain red shift or blue shift, the band shape and sign of each band are obviously consistent with the experimental spectrum.

In the figure, the positive Cotton effect is formed in the wavelength range of 350~500 nm, and the molecule O9H shows a significant red shift at a wavelength of 440.27 nm. The negative Cotton effect is generally formed in the short wavelength region of 250~350 nm. In the ECD spectrum, the chirality of the molecule was seen to be reversed near 270.14 nm. Chiral inversion occurs in this wavelength range. It can be seen from Figure 6 that the intramolecular electric field will affect the chirality of the molecule under the molecular twisted structure. When the molecular chirality changes, the magnetic dipole transition density of the twisted structure of the OO8H and OO9H polymers also changes (Figure 7), the S_5_ of OO8H has a positive absorption peak near 310 nm of the ECD spectrum (Figure 8), and the S_3_ of OO7H and the S_5_ of OO9H have opposite ECD peaks. 

## 3. Method

Gaussian 16 software [44], combined with density functional theory (DFT) [45], was used for this research. Based on the def2-SVP [46,47,48] basis functions, the molecular geometry of helicene for skeletal reconstruction in the solvent CH_2_Cl_2_ was investigated and analyzed. Then, based on the reconstructed structure, the time-dependent density functional theory (TDDFT) [49] was combined with the CAM-B3LYP functional [50] and def2-SVP basis functions for excited state calculations. Finally, orbital analysis, electron-hole pair analysis, and analysis of electrostatic potential, transition electric dipole moment, and transition magnetic dipole moment density were performed by using the Multiwfn3.8 program [51]. Three-dimensional spatial density maps were drawn using VMD software [52].

## 4. Conclusions

The geometric structure of nanographene can be changed, causing $P_{z}$ orbit polarization that creates an intramolecular electric field within the nanographene. The charge transfer-mediated photonic optical absorption behaviors and the electric and magnetic dipole moment-mediated chiral optical absorption behaviors can be effectively regulated by the intramolecular electric field, thereby changing the absorption spectrum and ECD spectrum of the molecules. In this work, a theoretical model is used to precisely control the spectra and chiral behavior of materials by twisting the geometric structure of nanographene to generate an intramolecular electric field. This work demonstrates the structure–activity relationship between the molecular twisted structure and spectra. This relationship can enable precise regulation of optical materials through intramolecular electric fields, which could be used for future applications of nanographene in solar cells, optoelectronics, catalysis, bioaggregation-induced luminescence, and other fields. 

## Figures and Tables

**Figure 1 molecules-28-01801-f001:**
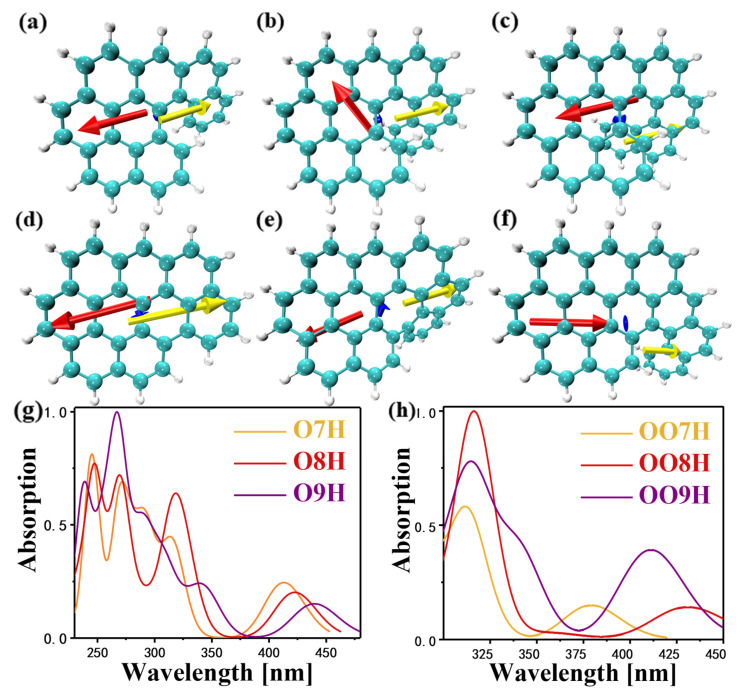
The pattern of dipole moments corresponding to 6 molecules under normalized conditions: (**a**) O7H, (**b**) O8H, (**c**) O9H, (**d**) OO7H, (**e**) OO8H, and (**f**) OO9H. The blue arrow is the direction of the overall dipole moment, the red arrow is the direction of the dipole moment of the benzopyrene fragment, and the yellow arrow is the direction of the dipole moment of the remaining helicene fragments. Comparison of the ultraviolet–visible absorption spectroscopy (UV–Vis) absorption spectra of molecules, (**g**): O7H, O8H, and O9H. (**h**): OO7H, OO8H, and OO9H.

**Figure 2 molecules-28-01801-f002:**
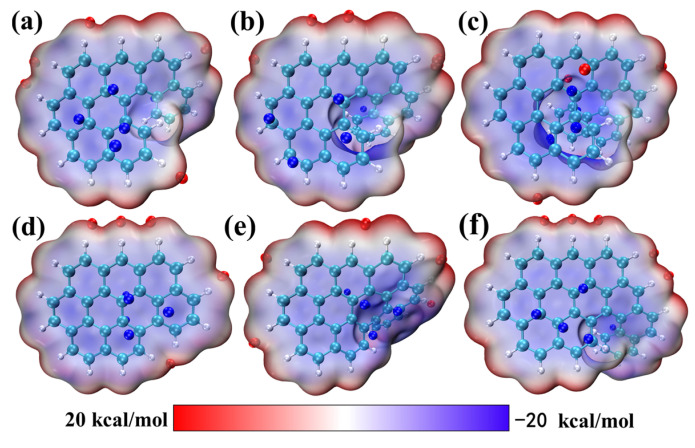
Electrostatic potential maps of 6 molecules: (**a**) O7H, (**b**) O8H, (**c**) O9H, (**d**) OO7H, (**e**) OO8H, and (**f**) OO9H.

**Figure 3 molecules-28-01801-f003:**
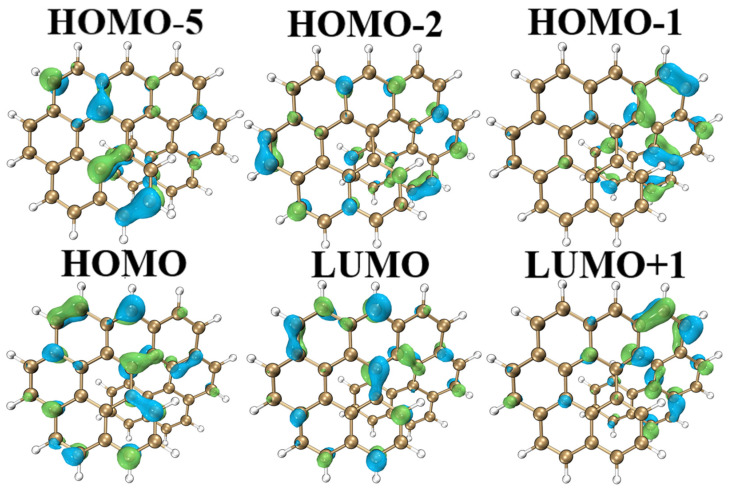
Schematic diagram of the molecular orbital of O9H.

**Figure 4 molecules-28-01801-f004:**
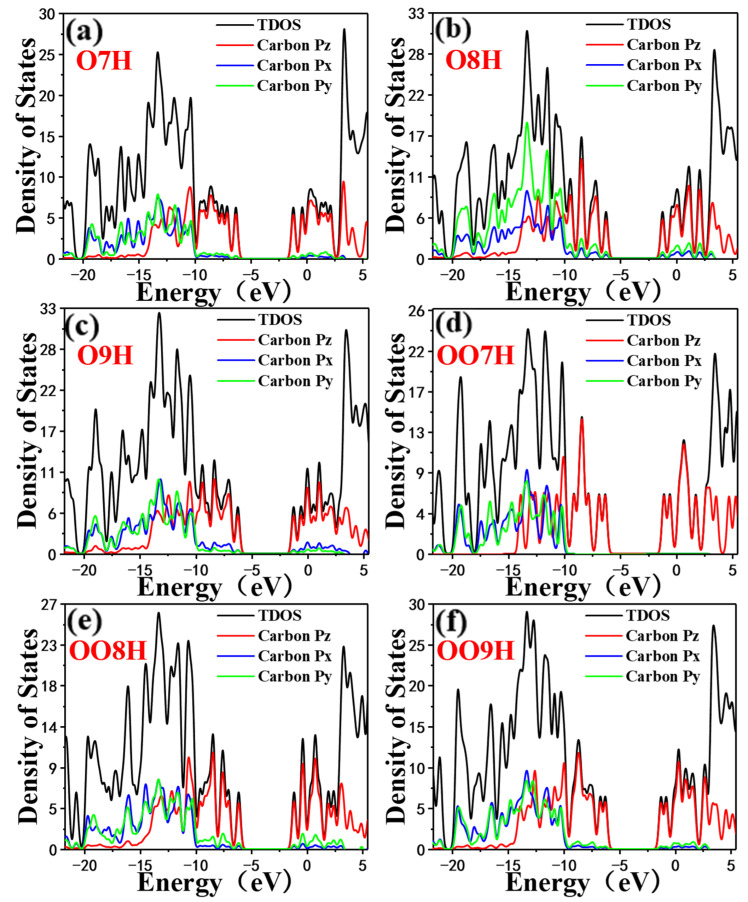
Density of states for the 6 molecules. The black curve is the total density of states, and the red, blue, and green curves are the local density of states on the $P_{z}$, $P_{x}$, and $P_{y}$ orbitals of carbon atoms.

**Figure 5 molecules-28-01801-f005:**
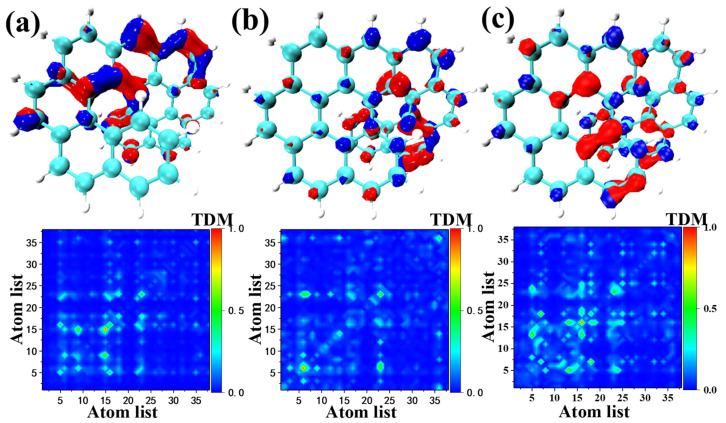
Electron-hole analysis diagram (upper) and transition density analysis diagram (lower) of O9H: S_1_ (**a**), S_6_ (**b**), and S_12_ (**c**).

**Figure 6 molecules-28-01801-f006:**
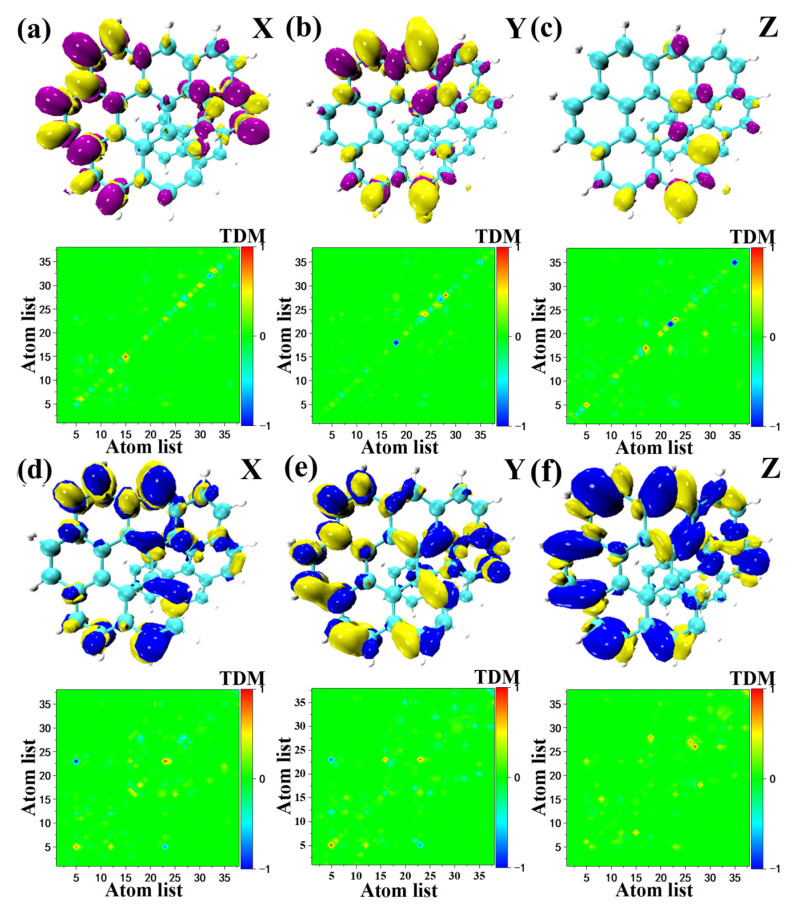
This is a figure. Schemes follow the same formatting. The Cartesian component isosurfaces of the transition electric dipole moment density (yellow and purple isosurfaces) and transition magnetic dipole moment density (yellow and blue isosurfaces) (upper part of (**a**–**c**)) and transition electric and magnetic couples (lower part of (**d**–**f**)) of S1 of the molecule O9H.

**Figure 7 molecules-28-01801-f007:**
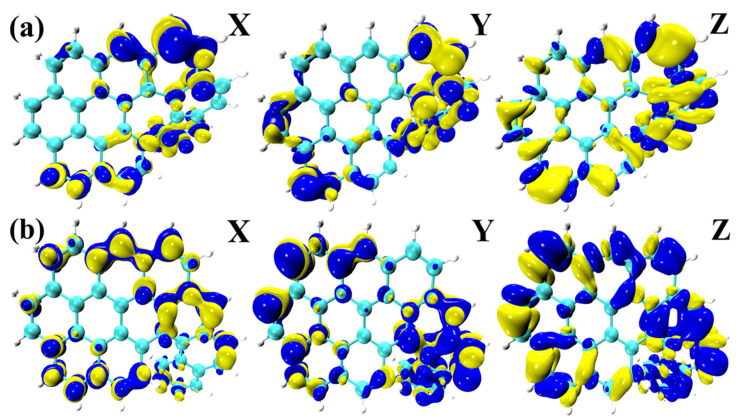
The Cartesian component isosurfaces of the transition magnetic dipole moment of the molecule (yellow and blue isosurfaces): (**a**) S_3_ of OO8H and (**b**) S_5_ of OO9H.

**Figure 8 molecules-28-01801-f008:**
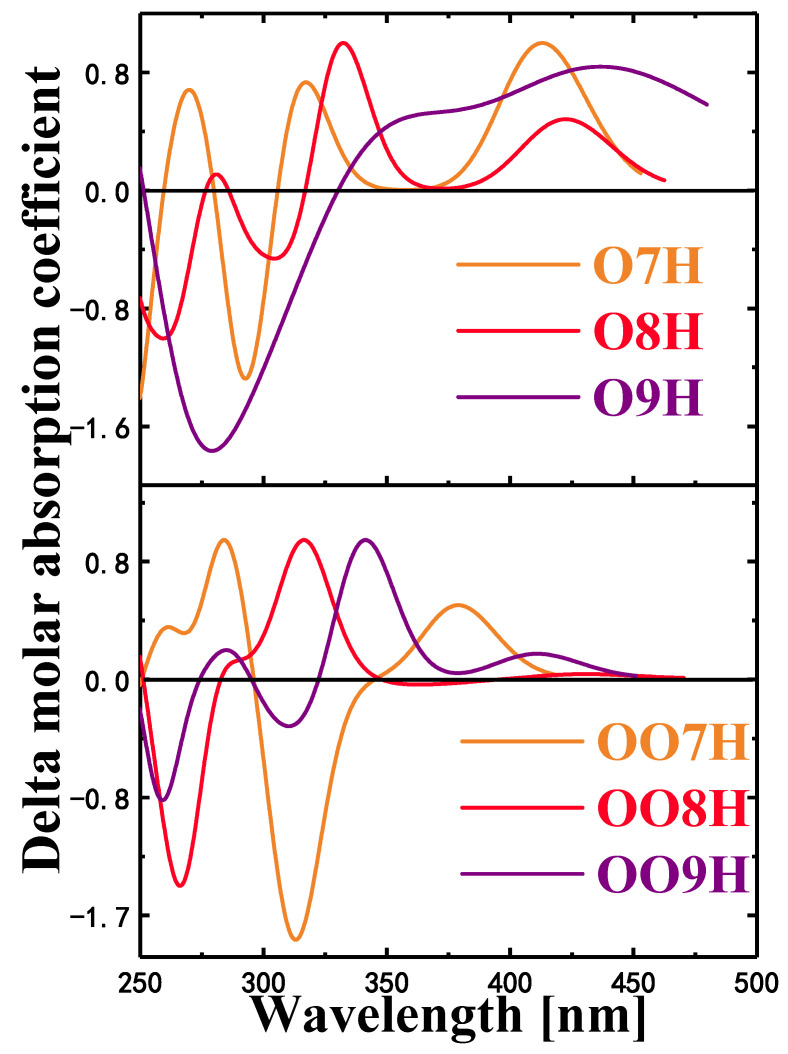
Electron circular dichroism combination diagram of O7H, O8H, and O9H (top), electron circular dichroism combination diagram of OO7H, OO8H, and OO9H (bottom).

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
