# Peer review of "Orbital Polarization-Dependent Fragment Twist-Induced Intramolecular Electric-Field-Driven Charge Transfer"

_molecules, 2023, doi:10.3390/molecules28041801_

Round 1

Reviewer 1 Report

In this manuscript, Jingang Wang and co-workers have shown the disruption of the plane of the π-orbital due to fused aromatic hydrocarbons when it is twisted under the application of an electric field. This directly affects the distribution of electron throughout the molecule and hence have different dipole moments. The following comments and suggestions are given for improvements.

1. There are many English typos and it has to be polished throughout the manuscript.

2. The green arrow in Figure 1(a-f) is not clear.

3. The expression of line 145 is not the same as the figure you quoted.

4. The image quality of figures requires to be improved. The logo text is different in size and has shadows.

Reviewer 2 Report

Comments

In this manuscript, computational methods were used to demonstrate electronic structures and spectroscopic properties of twisted graphene. The theoretically analysis will be of some help to the reader’s research.

However, there are some questions to solve as follows.

Change the caption “Figure 1” to “Figure 6” on page 6.

The size balance of letters and numbers in figure 8 is not so good.

Do you have any idea on stability of the twisted graphene ? (Is it possible to synthesize and isolate ?)
